



# Brown carbon absorption in the red and near infrared spectral region

**A. Hoffer[1], A. Tóth[2], M. Pósfai[2], C.E. Chung[3], A. Gelencsér[1,2]**

[1] {MTA-PE Air Chemistry Research Group, Veszprém, P.O. Box 158, H-8201, Hungary}

[2] {Department of Earth and Environmental Sciences, University of Pannonia, Veszprém, P.O. Box 158, H-8201, Hungary}

[3] {Division of Atmospheric Sciences, Desert Research Institute, Reno, NV 89512, USA}

Correspondence to: A. Gelencsér (gelencs@almos.uni-pannon.hu) and C.E. Chung (Eddy.Chung@dri.edu).

## Abstract

Black carbon aerosols have been conventionally assumed to be the only light-absorbing carbonaceous particles in the red and near-infrared spectral regions of solar radiation in the atmosphere. Here we report that contrary to the conventional belief tar balls (a specific type of organic aerosol particles from biomass burning) do absorb red and near infrared radiation significantly. Tar balls were produced in a laboratory experiment and their chemical and optical properties were measured. The absorption of these particles in the range between 470 and 950 nm was measured with an aethalometer, which is widely used to measure aerosol absorption in the field. We find that the absorption coefficient of tar balls at 880 nm exceeds 10% of that at 470 nm. This substantial absorption of red and infrared light is also evident from a relatively low Ångström coefficient (and a significant mass absorption coefficient) of tar balls between 470 and 950 nm. Retrievals of aerosol column optical properties from a global network of surface stations over vast tropical areas dominated by biomass burning suggest that tar balls are the predominant light-absorbing species of organic aerosols over acetone/methanol-soluble BrC or HULIS. Our results also infer that the role of BC (including Diesel soot) in global climate forcing has likely been overestimated at the expense of brown carbon (BrC) from biomass burning.



## 1 Introduction

In atmospheric science, black carbon (BC) aka soot aerosols have been conventionally assumed to be the only light-absorbing carbonaceous particles in the red and near-infrared

spectral regions of solar radiation in the atmosphere. Organic aerosols (OAs) are currently treated as either being weak absorbers of sunlight in the UV/blue region or having no solar absorption in radiation models (Myhre et al., 2013). Light-absorbing organic aerosols are also known as brown carbon (BrC) since they absorb blue light significantly but have practically zero absorption in the red band, yielding brownish colors (Andreae and Gelencsér, 2006). Separating

the BrC absorption from BC absorption in field and laboratory studies has relied on the assumption that no other carbonaceous particle type except BC absorbs solar radiation at the wavelength of ~700 nm or larger (Bahadur et al., 2012; Kirchstetter and Thatcher, 2012; Saleh et al., 2014; Lu et al., 2015). This common assumption has been used in spite of the finding by Alexander et al. (2008) who showed a sizable absorption by a specific class of BrC in the longer

wavelengths. Alexander et al. (2008) named these BrC particles "Brown Carbon Spheres" while indicating that the morphology of these particles is similar to that of tar balls. Alexander et al. (2008) derived the absorption using high spatial resolution electron energy-loss spectroscopy (EELS), which is not the prevailing method for absorption measurements.

The sources of atmospheric BrC are manifold, ranging from biomass burning emissions to

secondary formation in photochemical reactions yielding absorbing particles of various absorption efficiencies (Limbeck et al., 2003; Lukács et al., 2007). Tar balls are widespread in biomass burning smoke (Pósfai et al., 2004; Adachi and Buseck, 2011). These particles clearly belong to the class of BrC and not to BC, they are distinctly different from BC in morphology and other definition properties (Petzold et al., 2013). Tar balls are typically present as spherical

solid particles with diameters in the range of 25–500 nm, and can be readily identified by transmission electron microscopy (TEM-EDS), as against BC particles which always have fractal-like morphology. Both tar ball and BC particles are refractory as they can withstand the high vacuum and the irradiation by the electron beam in the TEM indefinitely. As far as elemental composition is concerned, fresh tar balls are nearly homogeneous mixtures of carbon

and oxygen at a molar ratio of about 10:1 as determined by TEM-EDS (Pósfai et al., 2004).



While BC particles are primarily formed in fossil fuel combustion and in the flaming stages of biomass fires, tar balls are often an abundant particle type in relatively aged smoke plumes from smoldering biomass fires (Pósfai et al., 2004).

Recently Tóth et al. (2014) have demonstrated that tar balls very similar to those observed in the atmosphere can be directly produced in the laboratory from liquid tar obtained by the dry distillation of wood. This lab experiment postulates that during biomass combustion direct ejection of liquid tar droplets from the pores followed by atmospheric aging generates tar ball particles in biomass smoke plumes. This allows us to measure the optical properties of tar balls directly without the interference of other combustion particles. Hoffer et al. (2016) measured the optical properties of tar balls generated with the above procedure, but they investigated the absorption parameters only up to 652 nm wavelength. In this study, similarly to the study by Tóth et al. (2014) and Hoffer et al. (2016) we generated tar balls in a laboratory from the dry distillate of hardwood (black locust) and softwood (Norway spruce) and measured the absorption properties up to 950 nm.

## 2  Experimental procedure

In this study tar balls were generated in an experimental setup similar to that used by Hoffer et al. (2016). Briefly, liquid tar was produced from dry distillation of 2 different wood types (*Robinia pseudoacacia* (black locust) and *Picea abies* (Norway spruce)) similar to that described in Tóth et al. (2014). The obtained liquid distillate consisted of an aqueous phase and an oily phase. During the experiments, only the aqueous phase was used. This phase was concentrated and taken up with methanol. Droplets were then generated from the methanol solution by an ultrasonic atomizer (1.6 MHz, Exo Terra Fogger, PT2080, Rolf C. Hagen Corp), and they were aged by heat at 650°C for about 1 second, using a tube furnace (Carbolite, MTF 10/25/130). During the particle generation the system was continually rinsed with $N_2$ containing 4% (v/v) $O_2$. The particles were then dried and diluted with dry filtered air. Before the optical measurements a PM1 cyclone (SCC 2.229) was deployed to remove the large particles (Dp> ~500 nm).

The absorption properties of the particles were measured with two optical instruments at different wavelengths. The absorption coefficient at 467 nm, 528 nm and 652 nm was measured with a continuous light absorption photometer (CLAP) with a time resolution of 5 seconds. In order to extend the measurements of the optical properties into the longer wavelength range (at



880 and 950 nm) an aethalometer (MAGEE AE42-7) was applied with a time resolution of 2 minutes. The CLAP data whose measurement principle is similar to that of PSAP are corrected according to Bond et al. (1999) and Ogren (2010) with the data processing algorithm applied by NOAA. The aethalometer data were corrected according to the Weingartner correction scheme

(Weingartner et al., 2003) and also by the Schmid correction (Schmid et al., 2006). The latter largely affects the Ångström exponent, whereas the former correction scheme has no effect on the Ångström exponent (Coen et al., 2010). In the correction we used the absorption coefficient measured by the CLAP at 528 nm as the reference value, keeping in mind that its reliability in the absolute scale is about 25% (Schmid et al., 2006).

During the experiment the size distribution between 7 and 800 nm was measured with a DMPS designed by the University of Helsinki.

The morphology and the elemental composition of the tar balls collected on TEM grids (lacey Formvar/carbon TEM copper grid of 200 mesh, Ted Pella Inc., USA) were studied in brightfield TEM images obtained using a Philips CM20 TEM operated at 200 kV accelerating

voltage. An ultra-thin-window Bruker Quantax X-ray detector was attached to the electron microscope that allowed the energy-dispersive X-ray analysis (EDS) of the elemental compositions of individual particles.

## 3 Results

### 3.1 Morphology and size distribution of generated tar balls

Figure 1 shows that the morphology of the generated particles is very similar to that of the freshly formed tar balls (Adachi and Buseck, 2011), as the particles are perfect or slightly distorted spheres. The size distribution measured with a DMPS was similar to that obtained previously by Hoffer et al. (2016). The volume size distribution consists of a double peak; the

larger is at 116 nm and 139 nm in the case of black locust and Norway spruce, respectively. In this mode, more than 96% of the particulate mass can be found. There is no difference in the molar C/O ratio between tar balls generated from the two wood types; it varies between 8.1 and 11.6 with an average of 9.3 as determined from TEM-EDS analyses.



### 3.2 Absorption properties of the generated tar balls

The absorption Ångström exponent (AAE) of the tar balls generated from the aqueous phase of the dry distillate from black locust and Norway spruce measured by the CLAP between 462 and 652 nm is 2.9 and 3.2, respectively. Similar values are obtained from the aethalometer data between 470 and 590 nm using the Weingartner correction scheme (Weingartner et al., 2003). The AAE in the same wavelength range for the black locust and Norway spruce calculated from the aethalometer data and corrected according the Schmid correction scheme (Schmid et al., 2006) was 3.2 and 3.5, respectively. The absorption measured at 370 nm by the aethalometer was, however, significantly lower than it would be predicted from AAE curve fitting. That is why the absorption data at this wavelength was omitted in the AAE calculations. Figure 2 shows that the AAE between 470 and 950 obtained by curve fitting is 3.1–3.2 and 3.4–3.6, depending on the applied correction scheme (Schmid correction gives higher values) for the black locust and Norway spruce, respectively. Note that the AAE is somewhat higher in the longer wavelength range (590–950 nm) than in the shorter range (470–590 nm). This difference is about a factor of 1.09–1.10 when the Weingartner correction is applied and 1.02–1.05 when the data are corrected according to Schmid. The error bars on figure 2 indicate the estimated 25% uncertainty of the absorption measurements for all wavelengths. Here we note that Chow et al., (2009) reported higher and wavelength dependent uncertainties between the filter based and photoacoustic methods (17-69%, greater differences at higher wavelengths), but the difference in AAE measured with different instruments were below 25%.

Since tar balls in the ambient atmosphere might have a size distribution different from those generated in our lab, the AAE of the atmospheric tar balls might be also somewhat different. The particle generation procedure used in the present study, as well as the measurement of the optical properties were similar to those used by Hoffer et al. (2016). These authors calculated the AAE for tar balls using an ambient size distribution determined by Pósfai et al. (2004) and found that the AAE decreased from ~2.9 to ~2.4 in the range between 462 and 652 nm. Since the size distribution of the generated particles in the present study was similar to that obtained by Hoffer et al. (2016) and the AAE of the tar balls in the same wavelength range (462–652 nm) were also similar, the AAE of freshly formed tar ball particles between 470 and 950 nm might be somewhat lower under ambient conditions than suggested by our calculations. On the other hand, Chow et al. (2009) showed that the AAE of ambient aerosol obtained from a photoacoustic



analyzer (PA) is noticeably higher than that obtained using filter-based instruments such as the aethalometer. In view of this, we propose that ambient freshly formed tar balls likely have an AAE of 2.7 ~ 3.6 in the wavelength range from 470 to 950 nm.

As Fig. 2 demonstrates, the absorption of tar balls is non-negligible in the near IR range. The
absorption coefficient at 880 nm exceeds by 10% that at 470 nm for both wood types, undermining the common assumption that all BrC particles have zero absorption at 880 nm. Even at 950 nm, the absorption coefficient is about 10% of that at 470 nm. The mass absorption efficiency of tar balls in the near IR range is expected to be substantial as well, given that the mass absorption efficiency of tar balls was estimated to be in the range 0.8–3 $m^2g^{-1}$ at 550 nm by
Hoffer et al. (2016). As Chow et al. (2009) demonstrated, absorption measured by filter-based instruments poses significant uncertainties and so follow-up studies are desired to reduce the uncertainty of estimated tar ball absorption.

### 3.3 Estimated contribution of tar balls to the absorption at K-puszta station

In this subsection, we estimate the contribution of the tar ball particles to the absorption at K-puszta station. During winter this station is affected by biomass burning used for domestic heating. Pósfai et al. (2004) investigated aerosol samples collected at this station and identified tar ball particles by electron microscopy. The concentration of levoglucosan is also elevated at the station during winter (Puxbaum et al., 2007).

In order to estimate the contribution of tar balls to the absorption measured at K-puszta station we calculated the absorption coefficient of a tar ball population with a Mie code at 652 nm and compared the calculated values with the measured absorption coefficient at this wavelength. The index of refraction of tar balls at 652 nm was taken from Hoffer et al. (2016). For the comparison we used the measured absorption coefficient (CLAP, at 652 nm) as well as
the measured particle number concentration (DMPS between 7 and 800 nm) at K-puszta during 10 days between 07.01.2014 and 27.01.2014. The shape of the size distribution of tar balls was taken from Pósfai et al. (2004) as it was determined from tar balls collected at K-puszta. Furthermore we assumed that the number share of tar ball particles is 20%, as Pósfai et al. (2004) reported that the contribution of tar ball particles to the total number concentration varies
between 0 and 40% in K-puszta. If we consider that only tar balls and soot are the absorbing components at 652 nm, the contribution of the tar balls to the absorption is 15–32%, on average



26%. Even if we consider that only 5% of the particles are tar balls, the contribution to the absorption is still 4–8% (on average 6%) at 652nm. This also indicates that the contribution of the tar balls to the absorption at higher wavelengths might be significant too, since the AAE of tar balls is close to that of the soot.

Here we note that the AAE measured at the station (1.58–1.88, on average 1.71 between 470 and 652 nm) during the investigated period (10 days, between 07.01.2014 and 27.01.2014) is very close to that (AAE=1.72 between 470 and 652 nm) obtained by the estimation assuming soot (AAE=1) and tar ball (AAE=3.15) as the only absorbing components, the latter causing 26% of the absorption at 652 nm. Since the contribution of humic like substances to the

absorption (few per cent at 550 nm, Hoffer et al., 2006) is incorporated in the measured AAE, the estimated contribution of the tar ball particles might be considered as an upper limit. If we assume that the contribution of tar ball particles to the total particle number concentration is only 5% (which also means that the contribution of tar balls to the absorption at 652 nm is 6%, see above), the calculated AAE, assuming soot and tar ball as the only absorbing components,

decreases significantly, it is 1.18 between 470 and 652 nm. Since the measured AAE is higher than the calculated value, the contribution of 6% to the absorption (5% in number concentration) can be considered as a lower value during the investigated period.

### 3.4  AERONET data analysis

Here, we analyze aerosol observations from a global network of well-calibrated sunphotometers (AERONET) (Holben et al., 1998). AERONET (Aerosol Robotic Network) data represent the column averages of aerosol optical properties. We used monthly AERONET AOD (Aerosol Optical Depth) and SSA (Single Scattering Albedo) level 2.0 data from 2001 to 2010. We selected biomass burning aerosol dominated data by choosing the data that have a SSA <

0.85 at 550 nm and have a scattering AOD Ångström Exponent > 1.7. The three wavelengths of 440, 675, 870 nm were used to compute Ångström exponents.

      The analysis of AERONET data shows that the AAE for biomass burning aerosols is near 1.15 in the range of 440 ~ 870 nm (Fig. 3), in accord with the results reported by Russell et al. (2010). The AAE is found to vary from 1.13 over 440 ~ 675 nm to 1.20 over 675 ~ 870 nm.

This variation of 1.13 ~ 1.20 is fairly small and suggests that acetone/methanol-soluble BrC or HULIS contributes very little to the total BrC absorption and tar balls contribute most, for the



following reason. Acetone/methanol-soluble BrC has a significant MAC (~0.1 m$^2$ g$^{-1}$ up to 0.6 m$^2$ g$^{-1}$ at 550 nm) and very high AAE values (from 6.0 to 11.0) at shorter wavelengths (Chen and Bond, 2010; Kirchstetter et al., 2004), while it has near-zero absorption at longer wavelengths (i.e., > 700 nm). Absorption of HULIS is also negligible above 700 nm (Hoffer et

al., 2006). Should BrC absorption be attributed to acetone/methanol-soluble species, the AAE of aerosols dominated by biomass burning would be markedly higher at shorter wavelengths (e.g., 440–675 nm) than in the red-infrared spectral region (e.g., 675–870 nm) over which BC would dominate. On the other hand, tar balls exhibit fairly uniform AAEs over the whole 440 ~ 870 nm range (Fig. 3) consequently show an insignificant AAE change across different wavelength

ranges. The small variation of 1.13 ~ 1.20 in AERONET AAE is well explained by combining tar balls and BC and leaving out acetone/methanol-soluble BrC.

AERONET-derived AAE should have uncertainties due to uncertainties in retrieval algorithms (Schuster et al., 2016). Also, there are a lot of uncertainties in filter-based absorption measurements (Chow et al., 2009), and so the wavelength dependence of MAC estimated for tar

balls and acetone/methanol-soluble BrC is subject to uncertainties. However, the aforementioned small variation of the AERONET AAE, even after considering all these uncertainties, is nearly impossible to explain with acetone/methanol-soluble BrC as the dominant BrC species. Thus, we conclude that in an atmospheric column the tar balls are the main BrC type in biomass burning aerosols.


## 4    Summary

We have used a CLAP to measure the absorption of tar ball particles between 462 and 652 nm and an aethalometer between 470 and 950 nm. The aethalometer has two measurement channels in the infrared region, 880 and 950 nm, thus allowing for direct measurement of the

light absorption in the red-infrared part of the spectrum. The Absorption Ångström Exponent (AAE) of tar balls over 470 ~ 950 nm is in the range between 2.7 and 3.6, but more importantly, the absorption coefficient at 880 nm exceeds 10% of that at 470 nm for both wood types (Fig. 2), clearly disproving the common assumption that all BrC particles have zero absorption at 880 nm. The determination of the contribution of BrC to aerosol absorption (Bahadur et al., 2012;

Kirchstetter and Thatcher, 2012; Saleh et al., 2013; Lu et al., 2015) has been based on the assumption that BrC has zero absorption at the wavelength of 700 nm or larger. The findings in





the present study have invalidated this common assumption. One of the resulting implications is that the role of BC, a significant fraction of which is derived from fossil fuel combustion (diesel soot), has likely been overestimated in global radiative forcing estimates if the biomass burning aerosol absorption in the red and near-infrared spectrum was attributed entirely to BC. The radiative and climatic impacts of tar balls need to be revised. We suggest that the global contribution of BrC to aerosol absorption is higher than previously estimated, pointing to a more significant role of organic aerosols in global radiative forcing and climate impacts.

**Acknowledgements**

10    The authors thank NOAA ESRL laboratory and the University of Helsinki for their support in data management and the size distribution measurements. This study was funded by the National Science Foundation (AGS-1455759).



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



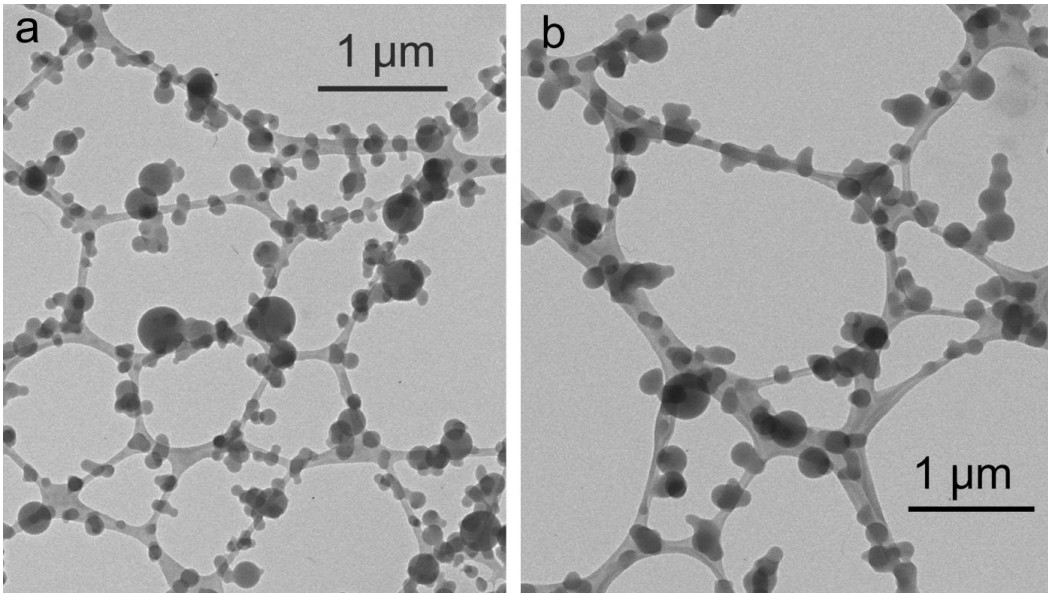

**Figure 1.** TEM images of tar balls generated from (**a**) black locust (*Robinia pseudoacacia*) and (**b**) Norway spruce (*Picea abies*).





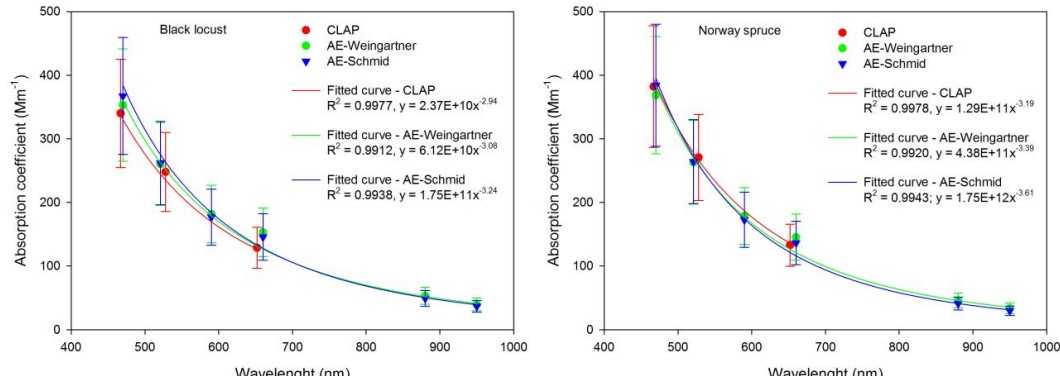

**Figure 2.** Absorption Ångström exponent of tar balls prepared from the liquid distillate of black locust and Norway spruce.




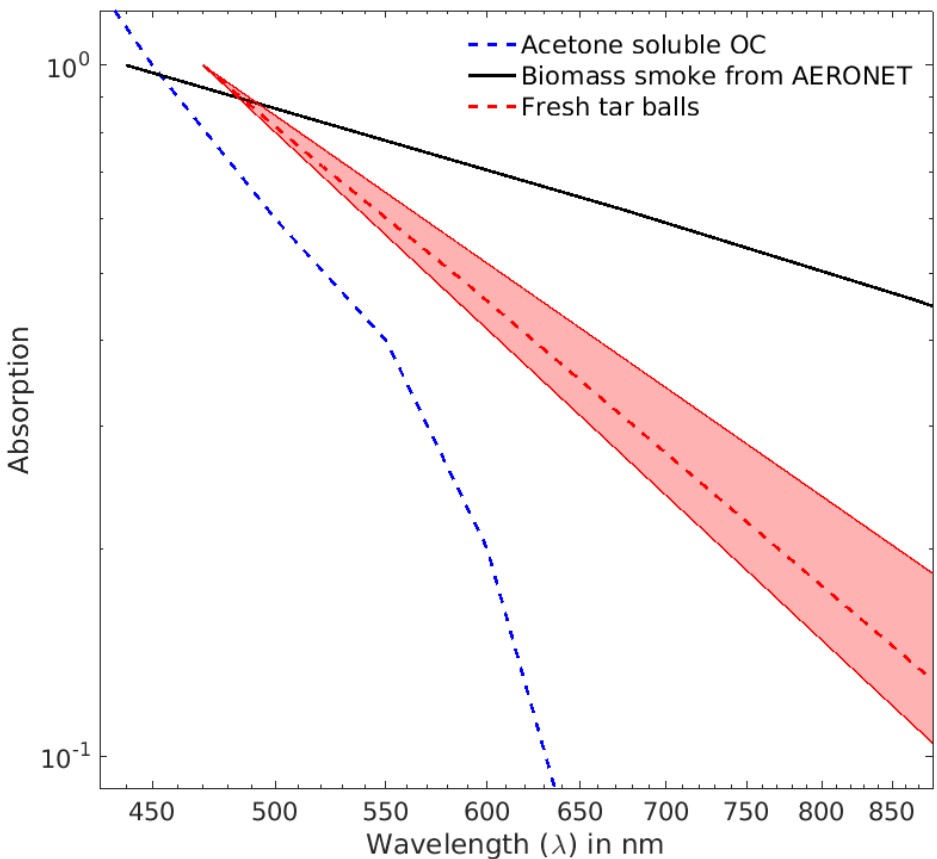

**Figure 3.** Normalized aerosol absorption. The absorption coefficient for each aerosol species is normalized by dividing the data by its value at or near 450 nm. The tar ball absorption shown here is the average of the measured values (red dashed line) and the estimated uncertainty (red solid lines). The average of the measured values and the uncertainty are obtained by averaging the measured data points after the Schmid and Weingartner corrections at each wavelength, and the upper and the lower bounds of the uncertainty correspond to AAE=3.6 and AAE=2.7, respectively. Note that tar balls we generated represent fresh tar balls and may not well represent tar balls transported over a long distance. Acetone-soluble OC absorption is from Kirchstetter et al. (2004).