# Peer review of "Brown carbon absorption in the red and near infrared spectral region"

_Atmospheric Chemistry and Physics, 2016_

## Short Comment (SC1) · 30 Jun 2016

My comments do not refer to the content of the manuscript, but to the means of presenting the data, specifically Figure 2, which presents the absorption coefficient against the wavelength on a linear-linear plot, but then fits the data to a power law to obtain an absorption Angstrom exponent. Were the data to be presented on a log-log plot, then the fits would be straight lines and it would be easier for the reader to judge the quality of the fits. Additionally the measurement at wavelength 370 nm could be included so that the reader could judge for him/herself how far from the fit it was.

Additionally, it was stated in the text that for Mie calculations, "the index of refraction of tar balls at 652 nm was taken from Hoffer et al. (2016)," but stating the value would save the reader from downloading another paper merely to find this value.

---

## Referee Comment (RC1) · Anonymous Referee #1 · 8 Jul 2016

In this paper the authors isolate components that comprise tar balls, re-aerosolize them and then measure the light absorption at multiple wavelengths. They find low (but non-trivial) levels of light absorption at high wavelengths. They make extensive use of Absorption Angstrom Exponents (AAEs) to interpret their data and to verify that indeed, according to them, tar balls are making significant contributions to overall absorption in the ambient atmosphere impacted by biomass burning.

The paper is interesting, but I do not believe the authors have presented a convincing argument. In my view the authors over interpret their data and make very strong statements based on weak evidence (eg, sections 3.3 and 3.4 are highly speculative). Yet, there is a simple and direct way to test their idea. They apparently have the technology to do it, since they have reported papers using it in the past. Measure the light absorption spectra, over a wide wavelength range, of the methanol solution containing

the dissolved components of tar balls.

The authors delineate tar balls from HULIS and BrC, implying it is a separate aerosol component in smoke. But it is generated by essentially dissolving an isolated product of smoke in methanol and then re-aerosolizing. Many studies, both from smoke generated in the lab and from collected ambient particles, have used methanol as a solvent to determine the light absorption spectra of smoke chromophores. Presumably these experiments/measurements would include chromophores that comprise tar balls. Thus I do not understand why tar balls are exclusive to BrC determined in methanol extracts. This is important because I am not aware of any strong evidence that methanol soluble smoke particles contain a significant number of chromophores that absorb at such high wavelengths (ie based on experiments where collected smoke particles are dissolved in methanol and the light absorption spectra measured). My interpretation is then, based on these observations, that what the authors are measuring is likely a very small component of the overall aerosol absorption spectra.

First, why not measure the light absorption spectra of the tar ball material they generate. Based on the generation method, they have a methanol solution from which they generate the aerosols. The chromophores must absorb at high wavelengths if the aerosols are also expected to. This is a simple and direct way to determine if the hypothesis is correct. It gets away from all the issues with correcting aethalometer data and the use of AAEs to infer absorption over the whole spectrum when only 5 or so wavelengths are measured. (Although it does have other limitations).

Secondly, if the authors are to claim that tar ball absorption in the red and infrared is important in the ambient atm, as noted above, why has no significant absorption been observed at these wavelengths based on filter methanol extracts in fresh and aged plumes? Again, there most be chromophores that absorb at the high wavelengths if this hypothesis is to be supported. These authors could obtain filters loaded with aged biomass burning smoke from their field site (K-puszta station), extract in methanol and measure the absorption spectra. If substantial absorption is observed in the red and

infrared, then they have direct proof. If they wish, use Mie theory to estimate the optical effects, based on some assumptions on size and mixing state.

Specific Comments;

The following statement is only partly true: Separating the BrC absorption from BC absorption in field and laboratory studies has relied on the assumption that no other carbonaceous particle type except BC absorbs solar radiation at the wavelength of ∼700 nm or larger (Bahadur et al., 2012; Kirchstetter and Thatcher, 2012; Saleh et al., 2014; Lu et al., 2015). All of the references are based on using optical instruments to measure light absorption. In that case, since total light absorption is measured, it is true that some assumptions have to be made about BC and BrC to separate the two. This statement is not true if one measures the light absorption of chromophores, what the authors refer to as acetone (or methanol) soluble OC.

Pg 6 it states: The absorption coefficient at 880 nm exceeds by 10% that at 470 nm for both wood types, undermining the common assumption that all BrC particles have zero absorption at 880 nm. As noted above, this assumption does not have to be made if one measures light absorption of the aerosol extract. The authors are ignoring much of the literature on BrC.

Regarding Fig. 3, it is claimed on page 8 lines 10 and 11 that the AERONET AAEs can be explained by tar balls and BC. But what are the assumptions about BC AAEs? State them. Given that there can be a range in BC properties (ie, effect of coatings on AAEs, etc), how can the authors make such a strong conclusion? It seems just the uncertainty in BC absorption could explain the AERONET data, no need for tar balls (ie, the AERONET AAEs are in the range that could be complete explained by BC). This needs to be discussed. Overall, I find this section highly speculative and not convincing given the well documented uncertainties in inferring AAEs from AERONET data, combined with uncertainties in BC AAEs.

---

## Short Comment (SC2) · 8 Jul 2016

Thanks for finding our paper interesting. We will revise the paper to fully address the issues raised by this reviewer. However, there appears to be a lot of misunderstanding about our paper. First, we believe that tar balls are a member of brown carbon (BrC) aerosols. We stated this and did not deny this in the paper.

Looks like this reviewer believes that methanol extracts contain all the brown carbon material (i.e., all the brown color chromophores). This thinking is shared by many others, and is what we are challenging through our paper. There is no solid evidence in the literature that methanol can extract all the BrC aerosols or organic aerosols by dissolution, and so it is strange that so many people believe this (i.e., that methanol extracts contain all the brown carbon). If there is any evidence, please let us know.

Like the reviewer doesn't believe, we don't believe either that methanol extracts absorb the far red and near-infrared spectrum. We are however presenting evidence that a) methanol can only extract a portion of brown color chromophores, and b) tar ball brown color chromosphere is not completely methanol soluble. We will improve the clarity during the revision.

---

## Referee Comment (RC2) · Anonymous Referee #2 · 16 Aug 2016

Review of acp-2016-452

GENERAL REMARKS

The manuscript presents measurements of the light absorption coefficient of laboratory-generated tar balls in the near infrared spectral region. The authors investigated the absorption Ångström exponent (AAE) of the tar balls and analysed field data from AERONET in order to prove their hypothesis that tar balls make up a major fraction of atmospheric brown carbon (BrC). Although the reported observation of a significant light absorption cross-section of tar balls in the investigated spectral region is of interest for the research field of climate effects of carbonaceous particles, the manuscript over-interprets the presented data and the drawn conclusions are not justified.

[Figure]

The study refers to a large extent to the work presented in a previous publication (Hoffer et al., 2016). In that paper, the method of aerosol generation is extensively described and the physical and optical properties of the tar balls are discussed in detail. Since the material presented here does not warrant publication as a full research paper, it is recommended to publish it as Comment or Technical Note with a strong link to the above-mentioned publication.

SPECIFIC COMMENTS

1| New material presented in the manuscript refer to the measurement of the aerosol light absorption coefficient in the near infrared spectrum, i.e., Section 3.2. This section, however, requires major revisions for the sake of clarity of the presented results. In particular the discussion of the AAE determination and error analysis requires more details. It is also recommended to follow the comment by E. Lewis in the discussion section and to present the analysis of the wavelength –dependence of the absorption coefficients as a log-log plot. Only this kind of data representation allows a statement whether or not the wavelength dependence of the light absorption properties can be described by a single exponent.

Then, the calculation of the AAE needs more details. Currently size distribution and refractive index values are taken from Hoffer et al. (2016). At least, the values of the refractive index and the size distribution have to be shown in this manuscript. Furthermore, the section needs to be written in a more quantitative way. Currently, key statements, e.g., on the AAE (page 6, line 2) or on the mass absorption coefficient (MAC; page 6, line 9) are presented as "we propose, that..." and "was estimated".

2| The section on the contribution of tar balls to the absorption at K-puszta station (Sec. 3.3) is speculative and confusing. When deducing the contribution of tar balls to atmospheric absorption from a comparison of Mie calculation and observations, a careful consideration of uncertainties from models and observations is needed for assessing the statistical significance of the results. In the given form, this is not possible. Concerning the separation of light absorption by black carbon (BC) and by tar balls via the AAE approach, it is confusing to read the conflicting statements that the AAE of tar balls is between 2.7 and 3.6 (page 6, line 2) and that the AAE of tar balls and soot is almost similar (page 7, lines 1-4). Recall that fresh BC has an AAE of approx. 1.0.

3| The section on the AERONET data analysis (Sec. 3.4) is also confusing. The authors do not describe the site, where the analysed AERONET data set is originating from. Then, they report an AAE of 1.15 for biomass burning events, which is at the lower limit of the values given by Russell et al. (2010) from AERONET data. Russel et al. report AAE values of 1.11 - 1.45 for biomass burning plumes and 1.05 to 1.12 for urban smoke plumes which are dominated by BC. The AAE results of 1.13 – 1.20 in AERONET data presented in this study can be easily explained by assuming a mixture of BC and BrC. If the authors decide to keep the AERONET part in the study, a detailed discussion of measurement uncertainties and statistical significance of presented results is required. Furthermore, the concluding statement that tar balls are the main BrC type in biomass burning has to be withdrawn, unless reasonable justification is presented.

REFERENCES

Hoffer, A., Tóth, A., NyirÅŚ-Kósa, I., Pósfai, M., and Gelencsér, A.: Light absorption properties of laboratory-generated tar ball particles, Atmos. Chem. Phys., 16, 239-246, doi: 10.5194/acp-16-239-2016, 2016.

Russell, P. B., Bergstrom, R. W., Shinozuka, Y., Clarke, A. D., DeCarlo, P. F., Jimenez, J. L., Livingston, J. M., Redemann, J., Dubovik, O., and Strawa, A.: Absorption Angstrom Exponent in AERONET and related data as an indicator of aerosol composition, Atmos. Chem. Phys., 10, 1155-1169, doi: 10.5194/acp-10-1155-2010, 2010.

---

## Author Comment (AC1) · 27 Sep 2016

1) In this paper the authors isolate components that comprise tar balls, re-aerosolize them and then measure the light absorption at multiple wavelengths. They find low (but non-trivial) levels of light absorption at high wavelengths. They make extensive use of Absorption Angstrom Exponents (AAEs) to interpret their data and to verify that indeed, according to them, tar balls are making significant contributions to overall absorption in the ambient atmosphere impacted by biomass burning.

We did not simply re-aerosolize previously isolated wood component (tar), but used a complex apparatus to simulate key processes that likely take place during wood combustion. The principle and the instrumental setup have been described in Tóth

et al. 2014. A key and indispensable step in the laboratory process was exposing the generated droplets to high temperature for a transient time period. It was shown previously that without this heat shock the droplets do not solidify, do not become rigid and refractory tar ball-like particles, and get distorted in shape upon impaction onto the sampling grid. Figure 4 in Hoffer et al. 2016 clearly demonstrate how the properties of the generated particles vary as a function of the applied temperature.

In order to emphasize the importance of this step the following sentence is added to the manuscript: This heat shock is of utmost importance as it strongly influences the chemical composition and optical properties of the formed particles (Hoffer et al., 2016).

2) The paper is interesting, but I do not believe the authors have presented a convincing argument. In my view the authors over interpret their data and make very strong statements based on weak evidence (eg, sections 3.3 and 3.4 are highly speculative). Yet, there is a simple and direct way to test their idea. They apparently have the technology to do it, since they have reported papers using it in the past. Measure the light absorption spectra, over a wide wavelength range, of the methanol solution containing the dissolved components of tar balls.

As has been stated above the tar ball particles were not simply produced by nebulization of methanol solution of the tar, but they underwent a transient heat shock which had a marked influence on both the optical and chemical properties of the generated particles. Therefore the suggested spectrometric measurement of dilute tar solution is not relevant concerning the optical properties of 'baked' solid tar ball particles.

3) The authors delineate tar balls from HULIS and BrC, implying it is a separate aerosol component in smoke. But it is generated by essentially dissolving an isolated product of smoke in methanol and then re-aerosolizing. Many studies, both from smoke generated in the lab and from collected ambient particles, have used methanol as a solvent to determine the light absorption spectra of smoke chromophores. Presumably these

experiments/measurements would include chromophores that comprise tar balls. Thus I do not understand why tar balls are exclusive to BrC determined in methanol extracts. This is important because I am not aware of any strong evidence that methanol soluble smoke particles contain a significant number of chromophores that absorb at such high wavelengths (ie based on experiments where collected smoke particles are dissolved in methanol and the light absorption spectra measured). My interpretation is then, based on these observations, that what the authors are measuring is likely a very small component of the overall aerosol absorption spectra.

The reviewer's arguments are based on the statement that the particles are generated by dissolving an isolated product of smoke. But in our experiments we have not used any component of smoke, instead we used liquid tar obtained by dry distillation of wood. It has to be noted that as against smoke particles in this case no flame chemistry was involved in the entire process. We have presented experimental evidence that 'baked' tar ball particles are practically insoluble in water or methanol. Therefore previous studies on methanol-soluble smoke chromophores were unable to measure the absorption properties of atmospheric tar balls.

4) First, why not measure the light absorption spectra of the tar ball material they generate. Based on the generation method, they have a methanol solution from which they generate the aerosols. The chromophores must absorb at high wavelengths if the aerosols are also expected to. This is a simple and direct way to determine if the hypothesis is correct. It gets away from all the issues with correcting aethalometer data and the use of AAEs to infer absorption over the whole spectrum when only 5 or so wavelengths are measured. (Although it does have other limitations).

See our comments above.

5) Secondly, if the authors are to claim that tar ball absorption in the red and infrared is important in the ambient atm, as noted above, why has no significant absorption been observed at these wavelengths based on filter methanol extracts in fresh and aged

plumes? Again, there most be chromophores that absorb at the high wavelengths if this hypothesis is to be supported. These authors could obtain filters loaded with aged biomass burning smoke from their field site (K-puszta station), extract in methanol and measure the absorption spectra. If substantial absorption is observed in the red and infrared, then they have direct proof. If they wish, use Mie theory to estimate the optical effects, based on some assumptions on size and mixing state.

See our comments above.

Specific Comments;

6) The following statement is only partly true: Separating the BrC absorption from BC absorption in field and laboratory studies has relied on the assumption that no other carbonaceous particle type except BC absorbs solar radiation at the wavelength of âĹij700 nm or larger (Bahadur et al., 2012; Kirchstetter and Thatcher, 2012; Saleh et al., 2014; Lu et al., 2015). All of the references are based on using optical instruments to measure light absorption. In that case, since total light absorption is measured, it is true that some assumptions have to be made about BC and BrC to separate the two. This statement is not true if one measures the light absorption of chromophores, what the authors refer to as acetone (or methanol) soluble OC.

The reviewer is right only if all BrC is assumed to be soluble in acetone (or methanol), which was shown not be the case with tar balls. See also our comments above.

7) Pg. 6 it states: The absorption coefficient at 880 nm exceeds by 10% that at 470 nm for both wood types, undermining the common assumption that all BrC particles have zero absorption at 880 nm. As noted above, this assumption does not have to be made if one measures light absorption of the aerosol extract. The authors are ignoring much of the literature on BrC.

Indeed, the common understanding is that BrC does not absorb in the IR range. Tar balls definitely belong to the class of BrC compounds since their key characteristics

markedly differ from the definition properties of BC (Petzold et al., 2013). Here we demonstrated experimentally that contrary to the common assumption (or belief) this specific type of BrC compounds – the tar balls – do show measurable absorption in the IR range of the solar spectrum. This fact was overlooked by previous BrC studies since tar balls are practically insoluble in acetone (or methanol).

8) Regarding Fig. 3, it is claimed on page 8 lines 10 and 11 that the AERONET AAEs can be explained by tar balls and BC. But what are the assumptions about BC AAEs? State them. Given that there can be a range in BC properties (ie, effect of coatings on AAEs, etc), how can the authors make such a strong conclusion? It seems just the uncertainty in BC absorption could explain the AERONET data, no need for tar balls (ie, the AERONET AAEs are in the range that could be complete explained by BC). This needs to be discussed. Overall, I find this section highly speculative and not convincing given the well documented uncertainties in inferring AAEs from AERONET data, combined with uncertainties in BC AAEs.

During the revision, we revised and expanded section 3.4, including the following paras: "We link a wavelength-independent AAE to the dominance of tar ball over BrC absorption, with an assumption that AERONET AAEs are accurate and BC AAE is quite wavelength independent. As for BC AAE, Gyawali et al. (2012) measured it with kerosene soot aerosols using a photoacoustic instrument (industry's benchmark instrument) and found a fairly wavelength-independent AAE of 0.8. On the other hand, AERONET-derived AAE should have uncertainties due to uncertainties in SSA retrieval algorithms (Schuster et al., 2016). It is thus possible that the AERONET SSA is overestimated at 440 nm compared to 675 and 870 nm and this creates spurious wavelength-independent AAE. The AERONET level 2 SSA data uncertainty was estimated to be about 0.03 and probably lower than 0.03 when AOD is large (Dubovik et al. 2000; Sayer et al. 2014). In an extremely unlikely scenario where all the selected AERONET SSA data in Fig. 4 have a positive bias of 0.03 at 440 nm and no bias at other wavelengths, the AAE over 440–675 nm would go up to 1.5. A hypothetical AAE variation

of 1.5 ∼ 1.20 is still small compared to the AAE analysis by Kirchstetter et al. (2004) who revealed the absorption properties of acetone soluble BrC for the first time. The biomass burning sample showed an AAE of 2.2 over the UV wavelengths and an AAE of 1.3 over 700∼1000 nm in Kirchstetter et al. (2004) (see Fig. 3 of their study). The AAEs in Kirchstetter et al. (2004) were derived by filter-based absorption measurement instruments. Filtering processes may alter particle shapes greatly (Subramanian et al. 2007), and particle shape affects aerosol absorption properties significantly. In addition, filter-based absorption measurements may have artifacts due to optical interactions between the concentrated particles themselves and the particles with the filter substrate (Moosmüller et al. 2009). Chow et al. (2009) indeed demonstrated that filter-based instruments even after various corrections give substantially lower AAE estimates than a photoacoustic instrument – instrument without filter. Thus, the relatively large AAE variation of 2.2 ∼ 1.3 in Kirchstetter et al. (2004) needs further validation. Nevertheless, a sharp AAE reduction from the UV wavelengths to the near infrared wavelengths in filter-based measurements has been taken qualitatively as the abundance of acetone-soluble BrC. The fact that AERONET AAE shows very little (or at least much smaller) AAE variation strongly suggests that tar balls contribute most to column average BrC absorption in biomass burning events."

Anonymous Referee #2

1) The manuscript presents measurements of the light absorption coefficient of laboratory-generated tar balls in the near infrared spectral region. The authors investigated the absorption Ångström exponent (AAE) of the tar balls and analysed field data from AERONET in order to prove their hypothesis that tar balls make up a major fraction of atmospheric brown carbon (BrC). Although the reported observation of a significant light absorption cross-section of tar balls in the investigated spectral region is of interest for the research field of climate effects of carbonaceous particles, the manuscript over-interprets the presented data and the drawn conclusions are not justified. The study refers to a large extent to the work presented in a previous publication (Hoffer et al., 2016). In that paper, the method of aerosol generation is extensively described and the physical and optical properties of the tar balls are discussed in detail. Since the material presented here does not warrant publication as a full research paper, it is recommended to publish it as Comment or Technical Note with a strong link to the above-mentioned publication.

In the previous publication (Hoffer et al., 2016) the authors did not measure the absorption experimentally above 652 nm, thus the absorption characteristics of tar balls in the IR region were not discussed in that paper. Indeed, in this study we used the very same method to generate tar ball particles in the laboratory as before, but the absorption characteristic of the tar balls were also measured directly with a 7-wavelength aethalometer at the wavelengths of 880 nm and 950 nm. This is the first direct experimental measurement of the IR absorption of 'pure' BrC-type particles (tar balls) that are abundant in biomass burning plumes but are definitely not BC. This undermines the common assumption in the literature that BrC does not absorb in the IR, which is generally based on spectrometric measurements of methanol (or acetone) solution of biomass smoke. (See also our answers to referee #1.) Therefore we believe that our new results deserve publication as a full research paper.

We added the following sentence to the end of the Introduction:

This is the first direct experimental measurement of the IR absorption of 'pure' BrC-type particles (tar balls) that are abundant in biomass burning plumes but are definitely not BC.

SPECIFIC COMMENTS

1) New material presented in the manuscript refer to the measurement of the aerosol light absorption coefficient in the near infrared spectrum, i.e., Section 3.2. This section, however, requires major revisions for the sake of clarity of the presented results. In particular the discussion of the AAE determination and error analysis requires more details. It is also recommended to follow the comment by E. Lewis in the discussion

section and to present the analysis of the wavelength –dependence of the absorption coefficients as a log-log plot. Only this kind of data representation allows a statement whether or not the wavelength dependence of the light absorption properties can be described by a single exponent. Then, the calculation of the AAE needs more details. Currently size distribution and refractive index values are taken from Hoffer et al. (2016). At least, the values of the refractive index and the size distribution have to be shown in this manuscript. Furthermore, the section needs to be written in a more quantitative way. Currently, key statements, e.g., on the AAE (page 6, line 2) or on the mass absorption coefficient (MAC; page 6, line 9) are presented as "we propose, that. . ." and "was estimated".

Section 3.2 has been modified, it is now written in a more quantitative way, explanatory text and numbers were added to increase the clarity of the section. The note about the difference in the AAE of laboratory generated tar balls measured in the longer and shorter range has been deleted for the sake of clarity. It might cause confusion since the R2 of the linear regression between the logarithm of the measured absorption coefficient and the logarithm of the wavelength between 470 nm and 950 nm is higher than 0.99.

The following sentences and texts were added to section 3.2:

page 5 line 17: This value was reported for a one-wavelength PSAP by Schmid et al., (2006).

page 5 line 22: since AAE depends on the size distribution as well

page 5, line 25: the ambient size distribution of these particles

page 5, line 30: (by about 20% based on the results by Hoffer et al., 2016)

page 6, line 1: (by 14–23%)

page 6, line 3: (The value 2.7 is the lowest AAE value of the generated tar balls from oak reported by Hoffer et al., 2016, whereas the upper value of the given range is the

highest AAE value obtained in the present study.)

Following the comment by E. Lewis Figure 2 (now Figure 3) has been modified.

The size distribution was not taken from Hoffer et al., 2016. Since size distribution affects the AAE, and the size distribution of the generated TBs was not exactly the same as that of the atmospheric tar balls, we noted that the measured AAE might be somewhat different than that of atmospheric tar balls. To evaluate the magnitude of this difference the results obtained by Hoffer et al. 2016 was taken. They found that the AAE decreased from ∼2.9 (the measured average value) to ∼2.4, which means that the AAE might be somewhat lower under ambient conditions than suggested in the present paper. The values of the index of refraction taken from Hoffer et al., 2016 are added into Section 3.3.

2) The section on the contribution of tar balls to the absorption at K-puszta station (Sec. 3.3) is speculative and confusing. When deducing the contribution of tar balls to atmospheric absorption from a comparison of Mie calculation and observations, a careful consideration of uncertainties from models and observations is needed for assessing the statistical significance of the results. In the given form, this is not possible. Concerning the separation of light absorption by black carbon (BC) and by tar balls via the AAE approach, it is confusing to read the conflicting statements that the AAE of tar balls is between 2.7 and 3.6 (page 6, line 2) and that the AAE of tar balls and soot is almost similar (page 7, lines 1-4). Recall that fresh BC has an AAE of approx. 1.0.

The objective section 3.3 is to assess the possible contribution of tar ball particles to the absorption at a given station (where the size distribution and the number share of tar balls were measured previously by TEM analysis (Pósfai et al., 2004)), based on the novel finding that absorption of these particles in the IR range is non-negligible. In this section we compared the measured absorption values with the ones calculated by the Mie theory for laboratory generated tar balls. The Mie calculation was also used previously in a backward direction to obtain the index of refraction of tar balls (and/or e.g.

HULIS particles). During these calculations the measured absorption and scattering values and the size distribution were implemented into the model calculations to obtain the values of the index of refraction at different wavelengths. In the present study we used the index of refraction and the observed size distribution and the number share of the particles to obtain the absorption coefficient of tar balls. These values were then compared with the measured absorption. It is known that absorption measurements are loaded with uncertainties, but even if we take these uncertainties into account the results indicate that the absorption of tar balls in the IR range may be non-negligible.

For the clarity the sentence has been modified. Our case study also implies that the contribution of the tar balls to the absorption at higher wavelengths might be non-negligible contrary to common assumption that it should be zero.

3) The section on the AERONET data analysis (Sec. 3.4) is also confusing. The authors do not describe the site, where the analysed AERONET data set is originating from. Then, they report an AAE of 1.15 for biomass burning events, which is at the lower limit of the values given by Russell et al. (2010) from AERONET data. Russel et al. report AAE values of 1.11 - 1.45 for biomass burning plumes and 1.05 to 1.12 for urban smoke plumes which are dominated by BC. The AAE results of 1.13 – 1.20 in AERONET data presented in this study can be easily explained by assuming a mixture of BC and BrC. If the authors decide to keep the AERONET part in the study, a detailed discussion of measurement uncertainties and statistical significance of presented results is required. Furthermore, the concluding statement that tar balls are the main BrC type in biomass burning has to be withdrawn, unless reasonable justification is presented.

We added Fig. 4B to show the selected AERONET data. Furthermore, we revised and expanded section 3.4 substantially to better clarify confusing points and better address measurement uncertainties. This is provided in our answer to the last comment of Referee #1.

Comments by E.Lewis

My comments do not refer to the content of the manuscript, but to the means of presenting the data, specifically Figure 2, which presents the absorption coefficient against the wavelength on a linear-linear plot, but then fits the data to a power law to obtain an absorption Angstrom exponent. Were the data to be presented on a log-log plot, then the fits would be straight lines and it would be easier for the reader to judge the quality of the fits. Additionally the measurement at wavelength 370 nm could be included so that the reader could judge for him/herself how far from the fit it was.

Figure 2 (now Figure 3) has been modified according to the suggestion

Additionally, it was stated in the text that for Mie calculations, "the index of refraction of tar balls at 652 nm was taken from Hoffer et al. (2016)," but stating the value would save the reader from downloading another paper merely to find this value.

The index of refraction value at 652 nm (1.82−0.15i) has been included in the text.